# Local Minimax Complexity of
# Stochastic Convex Optimization

**Yuancheng Zhu**
Wharton Statistics Department
University of Pennsylvania

**Sabyasachi Chatterjee**
Department of Statistics
University of Chicago

**John Duchi**
Department of Statistics
Department of Electrical Engineering
Stanford University

**John Lafferty**
Department of Statistics
Department of Computer Science
University of Chicago

## Abstract

We extend the traditional worst-case, minimax analysis of stochastic convex optimization by introducing a localized form of minimax complexity for individual functions. Our main result gives function-specific lower and upper bounds on the number of stochastic subgradient evaluations needed to optimize either the function or its "hardest local alternative" to a given numerical precision. The bounds are expressed in terms of a localized and computational analogue of the modulus of continuity that is central to statistical minimax analysis. We show how the computational modulus of continuity can be explicitly calculated in concrete cases, and relates to the curvature of the function at the optimum. We also prove a superefficiency result that demonstrates it is a meaningful benchmark, acting as a computational analogue of the Fisher information in statistical estimation. The nature and practical implications of the results are demonstrated in simulations.

## 1   Introduction

The traditional analysis of algorithms is based on a worst-case, minimax formulation. One studies the running time, measured in terms of the smallest number of arithmetic operations required by any algorithm to solve any instance in the family of problems under consideration. Classical worst-case complexity theory focuses on discrete problems. In the setting of convex optimization, where the problem instances require numerical rather than combinatorial optimization, Nemirovsky and Yudin [12] developed an approach to minimax analysis based on a first order oracle model of computation. In this model, an algorithm to minimize a convex function can make queries to a first-order "oracle," and the complexity is defined as the smallest error achievable using some specified minimum number of queries needed. Specifically, the oracle is queried with an input point $x \in \mathcal{C}$ from a convex domain $\mathcal{C}$, and returns an unbiased estimate of a subgradient vector to the function $f$ at $x$. After $T$ calls to the oracle, an algorithm $A$ returns a value $\widehat{x}_A \in \mathcal{C}$, which is a random variable due to the stochastic nature of the oracle, and possibly also due to randomness in the algorithm. The Nemirovski-Yudin analysis reveals that, in the worst case, the number of calls to the oracle required to drive the expected error $\mathbb{E}(f(\widehat{x}_A) - \inf_{x \in \mathcal{C}} f(x))$ below $\epsilon$ scales as $T = O(1/\epsilon)$ for the class of strongly convex functions, and as $T = O(1/\epsilon^2)$ for the class of Lipschitz convex functions.

In practice, one naturally finds that some functions are easier to optimize than others. Intuitively, if the function is "steep" near the optimum, then the subgradient may carry a great deal of information, and a stochastic gradient descent algorithm may converge relatively quickly. A minimax approach to analyzing the running time cannot take this into account for a particular function, as it treats the

worst-case behavior of the algorithm over all functions. It would be of considerable interest to be able to assess the complexity of solving an individual convex optimization problem. Doing so requires a break from traditional worst-case thinking.

In this paper we revisit the traditional view of the complexity of convex optimization from the point of view of a type of localized minimax complexity. In local minimax, our objective is to quantify the intrinsic difficulty of optimizing a specific convex function $f$. With the target $f$ fixed, we take an alternative function $g$ within the same function class $\mathcal{F}$, and evaluate how the maximum expected error decays with the number of calls to the oracle, for an optimal algorithm designed to optimize either $f$ or $g$. The local minimax complexity $R_T(f; \mathcal{F})$ is defined as the least favorable alternative $g$:

$$R_T(f; \mathcal{F}) = \sup_{g \in \mathcal{F}} \inf_{A \in \mathcal{A}_T} \max_{h \in \{f,g\}} \text{error}(A, h) \tag{1}$$

where $\text{error}(A, h)$ is some measure of error for the algorithm applied to function $h$. Note that here the the algorithm $A$ is allowed to depend on the function $f$ and the selected worst-case $g$. In contrast, the traditional global worst-case performance of the best algorithm, as defined by the minimax complexity $R_T(\mathcal{F})$ of Nemirovsky and Yudin, is

$$R_T(\mathcal{F}) = \inf_{A \in \mathcal{A}_T} \sup_{g \in \mathcal{F}} \text{error}(A, g). \tag{2}$$

The local minimax complexity can be thought of as the difficulty of optimizing the hardest alternative to the target function. Intuitively, a difficult alternative is a function $g$ for which querying the oracle with $g$ gives results similar to querying with $f$, but for which the value of $x \in \mathcal{C}$ that minimizes $g$ is far from the value that minimizes $f$.

Our analysis ties this function-specific notion of complexity to a localized and computational analogue of the modulus of continuity that is central to statistical minimax analysis [5, 6]. We show that the local minimax complexity gives a meaningful benchmark for quantifying the difficulty of optimizing a specific function by proving a superefficiency result; in particular, outperforming this benchmark at some function must lead to a larger error at some other function. Furthermore, we propose an adaptive algorithm in the one-dimensional case that is based on binary search, and show that this algorithm automatically achieves the local minimax complexity, up to a logarithmic factor. Our study of the algorithmic complexity of convex optimization is motivated by the work of Cai and Low [2], who propose an analogous definition in the setting of statistical estimation of a one-dimensional convex function. The present work can thus be seen as exposing a close connection between statistical estimation and numerical optimization of convex functions. In particular, our results imply that the local minimax complexity can be viewed as a computational analogue of Fisher information in classical statistical estimation.

In the following section we establish our notation, and give a technical overview of our main results, which characterize the local minimax complexity in terms of the computational modulus of continuity. In Section 2.2, we demonstrate the phenomenon of superefficiency of the local minimax complexity. In Section 3 we present the algorithm that adapts to the benchmark, together with an analysis of its theoretical properties. We also present simulations of the algorithm and comparisons to traditional stochastic gradient descent. Finally, we conclude with a brief review of related work and a discussion of future research directions suggested by our results.

## 2 Local minimax complexity

In this section, we first establish notation and define a modulus of continuity for a convex function $f$. We then state our main result, which links the local minimax complexity to this modulus of continuity.

Let $\mathcal{F}$ be the collection of Lipschitz convex functions defined on a compact convex set $\mathcal{C} \subset \mathbb{R}^d$. Given a function $f \in \mathcal{F}$, our goal is to find a minimum point, $x_f^* \in \arg\min_{x \in \mathcal{C}} f(x)$. However, our knowledge about $f$ can only be gained through a first-order oracle. The oracle, upon being queried with $x \in \mathcal{C}$, returns $f'(x) + \xi$, where $f'(x)$ is a subgradient of $f$ at $x$ and $\xi \sim \mathsf{N}(0, \sigma^2 I_d)$. When the oracle is queried with a non-differentiable point $x$ of $f$, instead of allowing the oracle to return an arbitrary subgradient at $x$, we assume that it has a deterministic mechanism for producing $f'(x)$. That is, when we query the oracle with $x$ twice, it should return two random vectors with the same mean $f'(x)$. Such an oracle can be realized, for example, by taking $f'(x) = \arg\min_{z \in \partial f(x)} \|z\|$. Here and throughout the paper, $\|\cdot\|$ denotes the Euclidean norm.

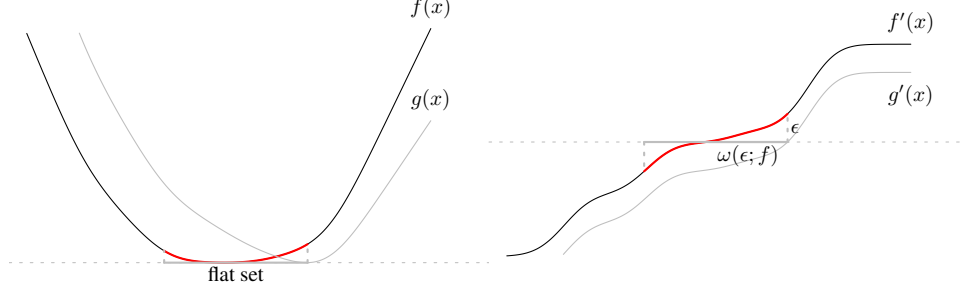

Figure 1: Illustration of the flat set and the modulus of continuity. Both the function $f$ (left) and its derivative $f'$ (right) are shown (black curves), along with one of the many possible alternatives, $g$ and its derivative $g'$ (solid gray curves), that achieve the sup in the definition of $\omega_f(\epsilon)$. The flat set contains all the points for which $|f'(x)| < \epsilon$, and $\omega_f(\epsilon)$ is the larger half width of the flat set.

Consider optimization algorithms that make a total of $T$ queries to this first-order oracle, and let $\mathcal{A}_T$ be the collection of all such algorithms. For $A \in \mathcal{A}_T$, denote by $\widehat{x}_A$ the output of the algorithm. We write $\mathrm{err}(x, f)$ for a measure of error for using $x$ as the estimate of the minimum point of $f \in \mathcal{F}$. In this notation, the usual minimax complexity is defined as

$$R_T(\mathcal{F}) = \inf_{A \in \mathcal{A}_T} \sup_{f \in \mathcal{F}} \mathbb{E}_f \, \mathrm{err}(\widehat{x}_A, f). \tag{3}$$

Note that the algorithm $A$ queries the oracle at up to $T$ points $x_t \in \mathcal{C}$ selected sequentially, and the output $\widehat{x}_A$ is thus a function of the entire sequence of random vectors $v_t \sim N(f'(x_t), \sigma^2 I_d)$ returned by the oracle. The expectation $\mathbb{E}_f$ denotes the average with respect to this randomness (and any additional randomness injected by the algorithm itself). The minimax risk $R_T(\mathcal{F})$ characterizes the hardness of the entire class $\mathcal{F}$. To quantify the difficulty of optimizing an individual function $f$, we consider the following local minimax complexity, comparing $f$ to its hardest local alternative

$$R_T(f; \mathcal{F}) = \sup_{g \in \mathcal{F}} \inf_{A \in \mathcal{A}_T} \max_{h \in \{f,g\}} \mathbb{E}_h \, \mathrm{err}(\widehat{x}_A, h). \tag{4}$$

We now proceed to define a computational modulus of continuity that characterizes the local minimax complexity. Let $\mathcal{X}_f^* = \arg\min_{x \in \mathcal{C}} f(x)$ be the set of minimum points of function $f$. We consider $\mathrm{err}(x, f) = \inf_{y \in \mathcal{X}_f^*} \|x - y\|$ as our measure of error. Define $d(f, g) = \inf_{x \in \mathcal{X}_f^*, y \in \mathcal{X}_g^*} \|x - y\|$ for $f, g \in \mathcal{F}$. It is easy to see that $\mathrm{err}(x, f)$ and $d(f, g)$ satisfy the *exclusion inequality*

$$\mathrm{err}(x, f) < \frac{1}{2} d(f, g) \quad \text{implies} \quad \mathrm{err}(x, g) \geq \frac{1}{2} d(f, g). \tag{5}$$

Next we define

$$\kappa(f, g) = \sup_{x \in \mathcal{C}} \|f'(x) - g'(x)\| \tag{6}$$

where $f'(x)$ is the unique subgradient of $f$ that is returned as the mean by the oracle when queried with $x$. For example, if we take $f'(x) = \arg\min_{z \in \partial f(x)} \|z\|$, we have

$$\kappa(f, g) = \sup_{x \in \mathcal{C}} \|\mathrm{Proj}_{\partial f(x)}(0) - \mathrm{Proj}_{\partial g(x)}(0)\| \tag{7}$$

where $\mathrm{Proj}_B(z)$ is the projection of $z$ to the set $B$. Thus, $d(f, g)$ measures the dissimilarity between two functions in terms of the distance between their minimizers, whereas $\kappa(f, g)$ measures the dissimilarity by the largest separation between their subgradients at any given point.

Given $d$ and $\kappa$, we define the *modulus of continuity* of $d$ with respect to $\kappa$ at the function $f$ by

$$\omega_f(\epsilon) = \sup \{ d(f, g) : g \in \mathcal{F}, \kappa(f, g) \leq \epsilon \}. \tag{8}$$

We now show how to calculate the modulus for some specific functions.

**Example 1.** Suppose that $f$ is a convex function on a one-dimensional interval $\mathcal{C} \subset \mathbb{R}$. If we take $f'(x) = \arg\min_{z \in \partial f(x)} \|z\|$, then

$$\omega_f(\epsilon) = \sup \left\{ \inf_{x \in \mathcal{X}_f^*} |x - y| : y \in \mathcal{C}, |f'(y)| < \epsilon \right\}. \tag{9}$$

The proof of this claim is given in the appendix. This result essentially says that the modulus of continuity measures the size (in fact, the larger half-width) of the the "flat set" where the magnitude of the subderivative is smaller than $\epsilon$. See Figure 1 for an illustration Thus, for the class of symmetric functions $f(x) = \frac{1}{k}|x|^k$ over $\mathcal{C} = [-1, 1]$, with $k > 1$,

$$\omega_f(\epsilon) = \epsilon^{\frac{1}{k-1}}. \tag{10}$$

For the asymmetric case $f(x) = \frac{1}{k_l}|x|^{k_l}I(-1 \leq x \leq 0) + \frac{1}{k_r}|x|^{k_r}I(0 < x \leq 1)$ with $k_l, k_r > 1$,

$$\omega_f(\epsilon) = \epsilon^{\frac{1}{k_l \vee k_r - 1}}. \tag{11}$$

That is, the size of the flat set depends on the flatter side of the function.

## 2.1 Local minimax is characterized by the modulus

We now state our main result linking the local minimax complexity to the modulus of continuity. We say that the modulus of the continuity has *polynomial growth* if there exists $\alpha > 0$ and $\epsilon_0$, such that for any $c \geq 1$ and $\epsilon \leq \epsilon_0/c$

$$\omega_f(c\epsilon) \leq c^\alpha \omega_f(\epsilon). \tag{12}$$

Our main result below shows that the modulus of continuity characterizes the local minimax complexity of optimization of a particular convex function, in a manner similar to how the modulus of continuity quantifies the (local) minimax risk in a statistical estimation setting [2, 5, 6], relating the objective to a geometric property of the function.

**Theorem 1.** *Suppose that $f \in \mathcal{F}$ and that $\omega_f(\epsilon)$ has polynomial growth. Then there exist constants $C_1$ and $C_2$ independent of $T$ and $T_0 > 0$ such that for all $T > T_0$*

$$C_1 \, \omega_f\left(\frac{\sigma}{\sqrt{T}}\right) \leq R_T(f; \mathcal{F}) \leq C_2 \, \omega_f\left(\frac{\sigma}{\sqrt{T}}\right). \tag{13}$$

**Remark 1.** We use the error metric $\text{err}(x, f) = \inf_{y \in \mathcal{X}_f^*} \|x - y\|$ here. For a given a pair $(\text{err}, d)$ that satisfies the exclusion inequality (5), our proof technique applies to yield the corresponding lower bound. For example, we could use $\text{err}(x, f) = \inf_{y \in \mathcal{X}_f^*} |v^T(x - y)|$ for some vector $v$. This error metric would be suitable when we wish to estimate $v^T x_f^*$, for example, the first coordinate of $x_f^*$. Another natural choice of error metric is $\text{err}(x, f) = f(x) - \inf_{x \in \mathcal{C}} f(x)$, with a corresponding distance $d(f, g) = \inf_{x \in \mathcal{C}} |f(x) - \inf_x f(x) + g(x) - \inf_x g(x)|$. For this case, while the proof of the lower bound stays exactly the same, further work is required for the upper bound, which is beyond the scope of this paper.

**Remark 2.** The results can be extended to oracles with more general noise models. In particular, the lower bounds will still hold with more general noise distributions, as long as Gaussian noise is a subclass. Indeed, in proving lower bounds assuming Gaussianity only makes solving the optimization problem easier. Our algorithm and upper bound analysis will go through for all sub-Gaussian noise oracles. For the ease of presentation, we will focus on Gaussian noise model for the current paper.

**Remark 3.** Although the theorem gives an upper bound for the local minimax complexity, this does not guarantee the existence of an algorithm that achieves the local complexity for any function. Therefore, it is important to design an algorithm that adapts to this benchmark for each individual function. We solve this problem in the one-dimensional case in Section 3.

The proof of this theorem is given in the appendix. We now illustrate the result with examples that verify the intuition that different functions should have different degrees of difficulty for stochastic convex optimization.

**Example 2.** For the function $f(x) = \frac{1}{k}|x|^k$ with $x \in [-1, 1]$ for $k > 1$, we have $R_T(f; \mathcal{F}) = O\left(T^{-\frac{1}{2(k-1)}}\right)$. This agrees with the minimax risk complexity for the class of Lipschitz convex functions that satisfy $f(x) - f(x_f^*) \geq \frac{\lambda}{2}\|x - x_f^*\|^k$ [14]. In particular, when $k = 2$, we recover the strongly convex case, where the (global) minimax complexity is $O\left(1/\sqrt{T}\right)$ with respect to the error $\text{err}(x, f) = \inf_{y \in \mathcal{X}_f^*} \|x - y\|$. We see a faster rate of convergence for $k < 2$. As $k \to \infty$, we also see that the error fails to decrease as $T$ gets large. This corresponds to the worst case for any Lipschitz convex function. In the asymmetric setting with $f(x) = \frac{1}{k_l}|x|^{k_l}I(-1 \leq x \leq 0) + \frac{1}{k_r}|x|^{k_r}I(0 < x \leq 1)$ with $k_l, k_r > 1$, we have $R_T(f; \mathcal{F}) = O\left(T^{-\frac{1}{2(k_l \vee k_r - 1)}}\right)$.

The following example illustrates that the local minimax complexity and modulus of continuity are consistent with known behavior of stochastic gradient descent for strongly convex functions.

**Example 3.** In this example we consider the error $\text{err}(x, f) = \inf_{y \in \mathcal{X}_f^*} |v^T(x - y)|$ for some vector $v$, and let $f$ be an arbitrary convex function satisfying $\nabla^2 f(x_f^*) \succ 0$ with Hessian continuous around $x_f^*$. Thus the optimizer $x_f^*$ is unique. If we define $g_w(x) = f(x) - w^T \nabla^2 f(x_f^*)x$, then $g_w(x)$ is a convex function with unique minimizer and

$$\kappa(f, g_w) = \sup_x \left\{ \left\| \nabla f(x) - (\nabla f(x) - \nabla^2 f(x_f^*)w) \right\| \right\} = \left\| \nabla^2 f(x_f^*)w \right\|. \tag{14}$$

Thus, defining $\delta(w) = x_f^* - x_{g_w}^*$,

$$\omega_f\left(\frac{\sigma}{\sqrt{T}}\right) \geq \sup_w \{|v^T\delta(w)| : \left\|\nabla^2 f(x_f^*)w\right\| \leq \sigma/\sqrt{T}\} \geq \sup_u \left| v^T \delta\left(\frac{\sigma}{\sqrt{T}}\nabla^2 f(x_f^*)^{-1}u\right) \right|. \tag{15}$$

By the convexity of $g_w$, we know that $x_{g_w}^*$ satisfies $\nabla f(x_{g_w}^*) - \nabla^2 f(x_f^*)^{-1}w = 0$, and therefore by the implicit function theorem, $x_{g_w}^* = x_f^* + w + o(\|w\|)$ as $w \to 0$. Thus,

$$\omega_f\left(\frac{\sigma}{\sqrt{T}}\right) \geq \frac{\sigma}{\sqrt{T}} \left\|\nabla^2 f(x_f^*)^{-1}v\right\| + o\left(\frac{\sigma}{\sqrt{T}}\right) \quad \text{as } T \to \infty. \tag{16}$$

In particular, we have the local minimax lower bound

$$\liminf_{T \to \infty} \sqrt{T} R_T(f; \mathcal{F}) \geq C_1 \sigma \left\|\nabla^2 f(x_f^*)^{-1}v\right\| \tag{17}$$

where $C_1$ is the same constant appearing in Theorem 1. This shows that the local minimax complexity captures the function-specific dependence on the constant in the strongly convex case. Stochastic gradient descent with averaging is known to adapt to this strong convexity constant [16, 13, 10]. Note that lower bounds of similar forms on the minimax complexity have been obtained in [11].

## 2.2 Superefficiency

Having characterized the local minimax complexity in terms of a computational modulus of continuity, we would now like to show that there are consequences to outperforming it at some function. This will strengthen the case that the local minimax complexity serves as a meaningful benchmark to quantify the difficulty of optimizing any particular convex function.

Suppose that $f$ is any one-dimensional function such that $\mathcal{X}_f^* = [x_l, x_r]$, which has as asymptotic expansion around $\{x_l, x_r\}$ of the form

$$f(x_l - \delta) = f(x_l) + \lambda_l \delta^{k_l} + o(\delta^{k_l}) \quad \text{and} \quad f(x_r + \delta) = f(x_r) + \lambda_r \delta^{k_r} + o(\delta^{k_r}) \tag{18}$$

for $\delta > 0$, some powers $k_l, k_r > 1$, and constants $\lambda_l, \lambda_r > 0$. The following result shows that if any algorithm significantly outperforms the local modulus of continuity on such a function, then it underperforms the modulus on a nearby function.

**Proposition 1.** *Let $f$ be any convex function satisfying the asymptotic expansion (21) around its optimum. Suppose that $A \in \mathcal{A}_T$ is any algorithm that satisfies*

$$\mathbb{E}_f \, \text{err}(\widehat{x}_A, f) \leq \sqrt{\mathbb{E}_f \, \text{err}(\widehat{x}_A, f)^2} \leq \delta_T \, \omega_f\left(\frac{\sigma}{\sqrt{T}}\right), \tag{19}$$

*where $\delta_T < C_1$. Define $g_{-1}(x) = f(x) - \epsilon_T x$ and $g_1(x) = f(x) + \epsilon_T x$, where $\epsilon_T$ is given by $\epsilon_T = \sqrt{\sigma^2 \log\left(\frac{C_1}{\delta_T}\right)/T}$. Then for some $g \in \{g_{-1}, g_1\}$, there exists $T_0$ such that $T \geq T_0$ implies*

$$\mathbb{E}_g \, \text{err}(\widehat{x}_A, g) \geq C \, \omega_g\left(\sqrt{\frac{\sigma^2 \log(C_1/\delta_T)}{T}}\right) \tag{20}$$

*for some constant $C$ that only depends on $k = k_l \vee k_r$.*

A proof of this result is given in the appendix, where it is derived as a consequence of a more general statement. We remark that while condition (19) involves the squared error $\sqrt{\mathbb{E}_f \operatorname{err}(\widehat{x}_A, f)^2}$, we expect that the result holds with only the weaker inequality on the absolute error $\mathbb{E}_f \operatorname{err}(\widehat{x}_A, f)$.

It follows from this proposition that if an algorithm $A$ significantly outperforms the local minimax complexity in the sense that (19) holds for some sequence $\delta_T \to 0$ with $\liminf_T e^T \delta_T = \infty$, then there exists a sequence of convex functions $g_T$ with $\kappa(f, g_T) \to 0$, such that

$$\liminf_{T \to \infty} \frac{\mathbb{E}_{g_T} \operatorname{err}(\widehat{x}_A, g_T)}{\omega_{g_T} \left( \sqrt{\sigma^2 \log\left(\frac{C_1}{\delta_T}\right)/T} \right)} > 0. \tag{21}$$

This is analogous to the phenomenon of superefficiency in classical parametric estimation problems, where outperforming the asymptotically optimal rate given by the Fisher information implies worse performance at some other point in the parameter space. In this sense, $\omega_f$ can be viewed as a computational analogue of Fisher information in the setting of convex optimization. We note that superefficiency has also been studied in nonparametric settings [1], and a similar result was shown by Cai and Low [2] for local minimax estimation of convex functions.

## 3 An adaptive optimization algorithm

In this section, we show that a simple stochastic binary search algorithm achieves the local minimax complexity in the one-dimensional case.

The general idea of the algorithm is as follows. Suppose that we are given a budget of $T$ queries to the oracle. We divide this budget into $T_0 = \lfloor T/E \rfloor$ queries over each of $E = \lfloor r \log T \rfloor$ many rounds, where $r > 0$ is a constant to be specified later. In each round, we query the oracle $T_0$ times for the derivative at the mid-point of the current interval. Estimating the derivative by averaging over the queries, we proceed to the left half of the interval if the estimated sign is positive, and to the right half of the interval if the estimated sign is negative. The details are given in Algorithm 1.

---
**Algorithm 1** Sign testing binary search

Input: $T$, $r$.
Initialize: $(a_0, b_0)$, $E = \lfloor r \log T \rfloor$, $T_0 = \lfloor T/E \rfloor$.
**for** $e = 1, \ldots, E$ **do**
  Query $x_e = (a_e + b_e)/2$ for $T_0$ times to get $Z_t^{(e)}$ for $t = 1, \ldots, T_0$.
  Calculate the average $\bar{Z}_{T_0}^{(e)} = \frac{1}{T_0} \sum_{t=1}^{T_0} Z_t^{(e)}$.
  If $\bar{Z}_{T_0}^{(e)} > 0$, set $(a_{e+1}, b_{e+1}) = (a_e, x_e)$.
  If $\bar{Z}_{T_0}^{(e)} \leq 0$, set $(a_{e+1}, b_{e+1}) = (x_e, b_e)$.
**end for**
Output: $x_E$.

---

We will show that this algorithm adapts to the local minimax complexity up to a logarithmic factor. First, the following result shows that the algorithm gets us close to the "flat set" of the function.

**Proposition 2.** *For $\delta \in (0, 1)$, let $C_\delta = \sigma \sqrt{2 \log(E/\delta)}$. Define*

$$\mathcal{I}_\delta = \left\{ y \in \operatorname{dom}(f) : |f'(y)| < \frac{C_\delta}{\sqrt{T_0}} \right\}. \tag{22}$$

*Suppose that $(a_0, b_0) \cap \mathcal{I}_\delta \neq \emptyset$. Then*

$$dist(x_E, \mathcal{I}_\delta) \leq 2^{-E}(b_0 - a_0) \tag{23}$$

*with probability at least $1 - \delta$.*

This proposition tells us that after $E$ rounds of bisection, we are at most a distance $2^{-E}(b_0 - a_0)$ from the flat set $\mathcal{I}_\delta$. In terms of the distance to the minimum point, we have

$$\inf_{x \in \mathcal{X}_f^*} |x_E - x| \leq 2^{-E}(b_0 - a_0) + \sup\left\{ \inf_{x \in \mathcal{X}_f^*} |x - y| : y \in \mathcal{I}_\delta \right\}. \tag{24}$$

If the modulus of continuity satisfies the polynomial growth condition, we then obtain the following.

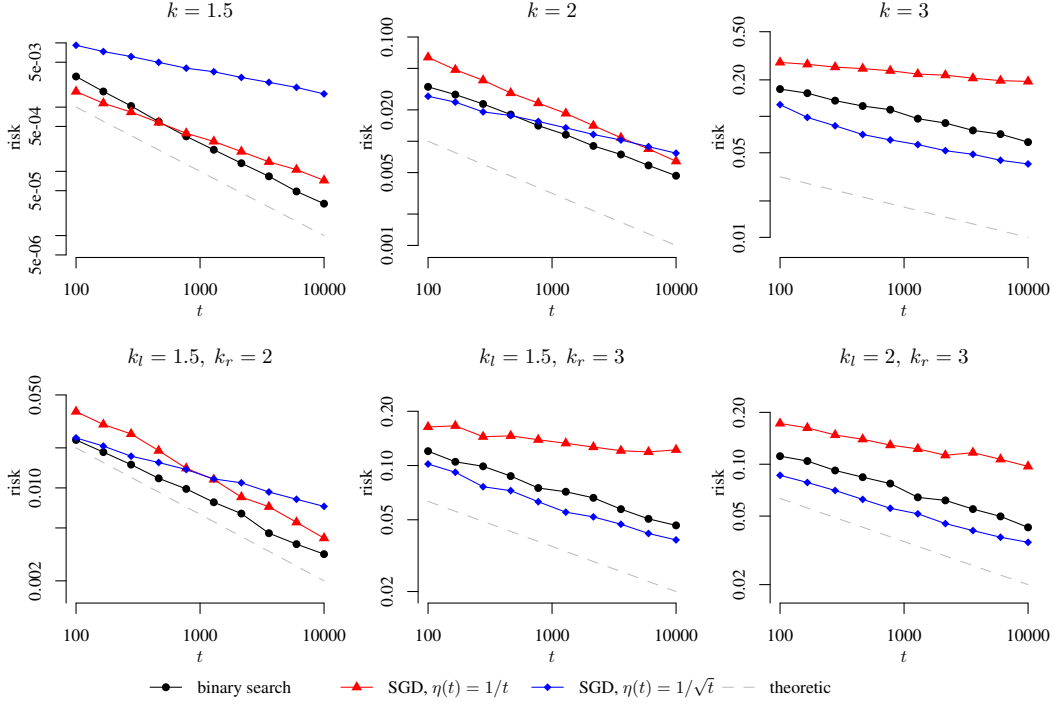

Figure 2: Simulation results: Averaged risk versus number of queries $T$. The black curves correspond to the risk of the stochastic binary search algorithm. The red and blue curves are for the stochastic gradient descent methods, red for stepsize $1/t$ and blue for $1/\sqrt{t}$. The dashed gray lines indicate the optimal convergence rate. Note that the plots are on a log-log scale. The plots on the top panels are for the symmetric cases $f(x) = \frac{1}{k}|x - x^*|^k$; the lower plots are for the asymmetric cases.

**Corollary 1.** *Let $\alpha_0 > 0$. Suppose $\omega_f$ satisfies the polynomial growth condition* (12) *with constant $\alpha \le \alpha_0$. Let $r = \frac{1}{2}\alpha_0$. Then with probability at least $1 - \delta$ and for large enough $T$,*

$$\inf_{x \in \mathcal{X}_f^*} |x_E - x| \le \widetilde{C}\omega_f\left(\frac{\sigma}{\sqrt{T}}\right) \tag{25}$$

*where the term $\widetilde{C}$ hides a dependence on $\log T$ and $\log(1/\delta)$.*

The proofs of these results are given in the appendix.

### 3.1 Simulations showing adaptation to the benchmark

We now demonstrate the performance of the stochastic binary search algorithm, making a comparision to stochastic gradient descent. For the stochastic gradient descent algorithm, we perform $T$ steps of update

$$x_{t+1} = x_t - \eta(t) \cdot \widehat{g}(x_t) \tag{26}$$

where $\eta(t)$ is a stepsize function, chosen as either $\eta(t) = \frac{1}{t}$ or $\eta(t) = \frac{1}{\sqrt{t}}$. We first consider the following setup with symmetric functions $f$:

1. The function to optimize is $f_k(x) = \frac{1}{k}|x - x^*|^k$ for $k = \frac{3}{2}$, 2 or 3.

2. The minimum point $x^* \sim \text{Unif}(-1, 1)$ is selected uniformaly at random over the interval.

3. The oracle returns the derivative at the query point with additive $N(0, \sigma^2)$ noise, $\sigma = 0.1$.

4. The optimization algorithms know *a priori* that the minimum point is inside the interval $(-2, 2)$. Therefore, the binary search starts with interval $(-2, 2)$ and the stochastic gradient descent starts at $x_0 \sim \text{Unif}(-2, 2)$ and project the query points to the interval $(-2, 2)$.

5. We carry out the simulation for values of $T$ on a logarithmic grid between 100 and 10,000. For each setup, we average the error $|\widehat{x} - x^*|$ over 1,000 runs.

The simulation results are shown in the top 3 panels of Figure 2. Several properties predicted by our theory are apparent from the simulations. First, the risk curves for the stochastic binary search algorithm parallel the gray curves. This indicates that the optimal rate of convergence is achieved. Thus, the stochastic binary search adapts to the curvature of different functions and yields the optimal local minimax complexity, as given by our benchmark. Second, the stochastic gradient descent algorithms with stepsize $1/t$ achieve the optimal rate when $k = 2$, but not when $k = 3$; with stepsize $1/\sqrt{t}$ SGD gets close to the optimal rate when $k = 3$, but not when $k = 2$. Neither leads to the faster rate when $k = \frac{3}{2}$. This is as expected, since the stepsize needs to be adapted to the curvature at the optimum in order to achieve the optimal rate.

Next, we consider a set of asymmetric functions. Using the same setup as in the symmetric case, we consider the functions of the form $f(x) = \frac{1}{k_l}|x - x^*|^{k_l} I(x - x^* \leq 0) + \frac{1}{k_r}|x - x^*|^{k_r} I(x - x^* > 0)$, for exponent pairs $(k_1, k_2)$ chosen to be $(\frac{3}{2}, 2)$, $(\frac{3}{2}, 3)$ and $(2, 3)$. The simulation results are shown in the bottom three panels of Figure 2. We observe that the stochastic binary search once again achieves the optimal rate, which is determined by the flatter side of the function, that is, the larger of $k_l$ and $k_r$.

## 4 Related work and future directions

In related recent work, Ramdas and Singh [14] study minimax complexity for the class of Lipschitz convex functions that satisfy $f(x) - f(x_f^*) \geq \frac{\lambda}{2}\|x - x_f^*\|^k$. They show that the minimax complexity under the function value error is of the order $T^{-\frac{k}{2(k-1)}}$. Juditski and Nesterov [8] also consider minimax complexity for the class of $k$-uniformly convex functions for $k > 2$. They give an adaptive algorithm based on stochastic gradient descent that achieves the minimax complexity up to a logarithmic factor. Connections with active learning are developed in [15], with related ideas appearing in [3]. Adaptivity in this line of work corresponds to the standard notion in statistical estimation, which seeks to adapt to a large subclass of a parameter space. In contrast, the results in the current paper quantify the difficulty of stochastic convex optimization at a much finer scale, as the benchmark is determined by the specific function to be optimized.

The stochastic binary search algorithm presented in Section 3, despite being adaptive, has a few drawbacks. It requires the modulus of continuity of the function to satisfy polynomial growth, with a parameter $\alpha$ bounded away from 0. This rules out cases such as $f(x) = |x|$, which should have an error that decays exponentially in $T$; it is of interest to handle this case as well. It would also be of interest to construct adaptive optimization procedures tuned to a fixed numerical precision. Such procedures should have different running times depending on the hardness of the problem. Progress on both problems has been made, and will be reported elsewhere.

Another challenge is to remove the logarithmic factors appearing in the binary search algorithm developed in Section 3. In one dimension, stochastic convex optimization is intimately related to a noisy root finding problem for a monotone function taking values in $[-a, a]$ for some $a > 0$. Karp and Kleinberg [9] study optimal algorithms for such root finding problems in a discrete setting. A binary search algorithm that allows backtracking is proposed, which saves log factors in the running time. It would be interesting to study the use of such techniques in our setting.

Other areas that warrant study involve the dependence on dimension. The scaling with dimension of the local minimax complexity and modulus of continuity is not fully revealed by the current analysis. Moreover, the superefficiency result and the adaptive algorithm presented here are only for the one-dimensional case. We note that a form of adaptive stochastic gradient algorithm for the class of uniformly convex functions in general, fixed dimension is developed in [8].

Finally, a more open-ended direction is to consider larger classes of stochastic optimization problems. For instance, minimax results are known for functions of the form $f(x) := \mathbb{E}\, F(x; \xi)$ where $\xi$ is a random variable and $x \mapsto F(x; \xi)$ is convex for any $\xi$, when $f$ is twice continuously differentiable around the minimum point with positive definite Hessian. However, the role of the local geometry is not well understood. It would be interesting to further develop the local complexity techniques introduced in the current paper, to gain insight into the geometric structure of more general stochastic optimization problems.

## Acknowledgments

Research supported in part by ONR grant 11896509 and NSF grant DMS-1513594. The authors thank Tony Cai, Praneeth Netrapalli, Rob Nowak, Aaron Sidford, and Steve Wright for insightful discussions and valuable comments on this work.

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
