[Supplementary Material]

# A Proof of claim in Example 1

*Proof.* We will show for a convex function $f : \mathcal{C} \to \mathbb{R}$ with a compact and convex domain $\mathcal{C} \subset \mathbb{R}$

$$\omega_f(\epsilon) = \sup \left\{ \inf_{x \in \mathcal{X}_f^*} |x - y| : y \in \mathcal{C}, |f'(y)| < \epsilon \right\}. \tag{27}$$

To ease notation, we denote the quantity on the right hand side by $\delta_f(\varepsilon)$.

As a first step we show $\omega_f(\varepsilon) \leq \delta_f(\varepsilon)$. It suffices to show that for any $g \in \mathcal{F}$ satisfying $\kappa(f,g) \leq \varepsilon$, we have $d(f,g) \leq \delta_f(\varepsilon)$. Recalling the definition of $\kappa(f,g) \leq \varepsilon$, we have for any $x \in \mathcal{C}$, $|f'(x) - g'(x)| \leq \varepsilon$. Take $x \in \mathcal{X}_g^*$ to be a minimizer of $g$, and we have $|f'(x)| \leq \varepsilon$. Therefore, $\mathcal{X}_g^* \subseteq \{x : |f'(x)| \leq \varepsilon\}$. Let's now consider two cases. If $\mathcal{X}_g^* \cap \{x : |f'(x)| < \varepsilon\} \neq \emptyset$, then

$$d(f,g) = \inf_{x \in \mathcal{X}_f^*, \, y \in \mathcal{X}_g^*} |x - y| \tag{28}$$

$$\leq \inf_{y \in \mathcal{X}_g^* \cap \{z : |f'(z)| < \varepsilon\}} \inf_{x \in \mathcal{X}_f^*} |x - y| \tag{29}$$

$$\leq \sup_{y \in \{z : |f'(z)| < \varepsilon\}} \inf_{x \in \mathcal{X}_f^*} |x - y| \tag{30}$$

$$= \delta_f(\varepsilon). \tag{31}$$

Now assume that $\mathcal{X}_g^* \cap \{x : |f'(x)| < \varepsilon\} = \emptyset$ and this means that $\mathcal{X}_g^* \subseteq \{x : |f'(x)| = \varepsilon\}$. Without lost of generality, we further assume that $\mathcal{X}_g^* \subseteq \{x : f'(x) = \varepsilon\}$. Since $\kappa(f,g) \leq \varepsilon$, we must have $\inf \mathcal{X}_g^* = \inf\{x : f'(x) = \varepsilon\} = \sup\{x : f'(x) < \varepsilon\}$ and therefore

$$d(f,g) = \inf_{y \in \mathcal{X}_g^*} \inf_{x \in \mathcal{X}_f^*} |x - y| \leq \sup_{y \in \{z : |f'(z)| < \varepsilon\}} \inf_{x \in \mathcal{X}_f^*} |x - y| = \delta_f(\varepsilon). \tag{32}$$

This concludes the first part of the proof as we have shown $\omega_f(\varepsilon) \leq \delta_f(\varepsilon)$.

Next, we are going to prove $\omega_f(\varepsilon) \geq \delta_f(\varepsilon)$. Without loss of generality, we assume the larger half of the "flat set" is to the right of the minimum point, that is

$$\delta_f(\varepsilon) = \sup \left\{ \inf_{x \in \mathcal{X}_f^*} |x - y| : y \in \mathcal{C}, 0 \leq f'(y) < \epsilon \right\}. \tag{33}$$

Then there exists a sequence of differentiable points of $f$, $\{y_i\}_{i=1}^{\infty}$, which satisfies $\lim_{i \to \infty} \inf_{x \in \mathcal{X}_f^*} |x - y_i| = \delta_f(\varepsilon)$ and $0 \leq f'(y_i) < \varepsilon$. Define a sequence of functions $\{g_i\}_{i=1}^{\infty}$ such that $g_i'(x) = f'(x) - \frac{f'(y_i) + \varepsilon}{2}$ at the differentiable point of $f$. It is easy to check that $g_i$ satisfies that $\kappa(f, g_i) \leq \varepsilon$. Also, $g'(y_i) = \frac{f'(y_i) - \varepsilon}{2} < 0$, so $\inf_{x \in \mathcal{X}_f^*, y \in \mathcal{X}_{g_i}^*} |x - y| \geq \inf_{x \in \mathcal{X}_f^*} |x - y_i|$. We have

$$\omega_f(\varepsilon) \geq \sup_{i=1,\dots,\infty} d(f, g_i) \geq \lim_{i \to \infty} \inf_{x \in \mathcal{X}_f^*, y \in \mathcal{X}_{g_i}^*} |x - y| \geq \lim_{i \to \infty} \inf_{x \in \mathcal{X}_f^*} |x - y_i| = \delta_f(\varepsilon). \tag{34}$$

Thus, we have shown $\omega_f(\varepsilon) \geq \delta_f(\varepsilon) \geq \omega_f(\varepsilon)$ and hence $\omega_f(\varepsilon) = \delta_f(\varepsilon)$. $\qquad\square$

# B Proof of Theorem 1

## B.1 Lower bound

For a function $f \in \mathcal{F}$, let $P_f$ denote the distribution of stochastic gradients observable by an estimation scheme $\widehat{x}$, and let $P_f^T$ denote the distribution of $T$ sequentially queried stochastic gradients for $f$. We define the pairwise minimax risk for optimization of a pair of function $f$ and $g$ by

$$R_T(f,g) := \inf_{A \in \mathcal{A}_T} \max \left\{ \mathbb{E}_f \, \mathrm{err}(\widehat{x}_A, f), \mathbb{E}_g \, \mathrm{err}(\widehat{x}_A, g) \right\}, \tag{35}$$

and the local minimax lower bound can be written as

$$R_T(f; \mathcal{F}) := \sup_{g \in \mathcal{F}} R_T(f, g). \tag{36}$$

Let us show how the modulus of continuity gives a lower bound. We first state a lemma.

**Lemma 1.** *Let $f, g$ be arbitrary convex functions and $d$ satisfy the exclusion inequality (5). Then*

$$R_T(f, g) \geq \frac{d(f, g)}{4} \left(1 - \left\| P_f^T - P_g^T \right\|_{\mathrm{TV}}\right). \tag{37}$$

*Proof.* Temporarily hiding the number of iterations $T$ for simplicity, we have by Markov's inequality that

$$\max \left\{ \mathbb{E}_f \, \mathrm{err}(\widehat{x}_A, f), \mathbb{E}_g \, \mathrm{err}(\widehat{x}_A, g) \right\} \tag{38}$$

$$\geq \frac{1}{2} d(f, g) \max \left\{ P_f(\mathrm{err}(\widehat{x}_A, f) \geq \frac{1}{2} d(f, g)), P_g(\mathrm{err}(\widehat{x}_A, g) \geq \frac{1}{2} d(f, g)) \right\}. \tag{39}$$

Now, we apply an essentially standard reduction of estimation to testing, because we have

$$2 \max \left\{ P_f(\mathrm{err}(\widehat{x}_A, f) \geq \frac{1}{2} d(f, g)), P_g(\mathrm{err}(\widehat{x}_A, g) \geq \frac{1}{2} d(f, g)) \right\} \tag{40}$$

$$\geq P_f(\mathrm{err}(\widehat{x}_A, f) \geq \frac{1}{2} d(f, g)) + P_g(\mathrm{err}(\widehat{x}_A, g) \geq \frac{1}{2} d(f, g)) \tag{41}$$

$$= 1 - P_f(\mathrm{err}(\widehat{x}_A, f) < \frac{1}{2} d(f, g)) + P_g(\mathrm{err}(\widehat{x}_A, g) \geq \frac{1}{2} d(f, g)) \tag{42}$$

$$\geq 1 - P_f(\mathrm{err}(\widehat{x}_A, g) \geq \frac{1}{2} d(f, g)) + P_g(\mathrm{err}(\widehat{x}_A, g) \geq \frac{1}{2} d(f, g)), \tag{43}$$

where in the last line we have used the exclusion inequality to see that $\mathrm{err}(\widehat{x}_A, f) < \frac{1}{2} d(f, g)$ implies $\mathrm{err}(\widehat{x}_A, g) \geq \frac{1}{2} d(f, g)$ so that

$$P_f(\mathrm{err}(\widehat{x}_A, f) < \frac{1}{2} d(f, g)) \leq P_f(\mathrm{err}(\widehat{x}_A, g) \geq \frac{1}{2} d(f, g)). \tag{44}$$

Thus, we find that

$$\frac{4}{d(f, g)} \max \left\{ \mathbb{E}_f \, \mathrm{err}(\widehat{x}_A, f), \mathbb{E}_g \, \mathrm{err}(\widehat{x}_A, g) \right\} \geq \inf_S \left\{ 1 - P_f^T(S) + P_g^T(S) \right\} = 1 - \left\| P_f^T - P_g^T \right\|_{\mathrm{TV}}, \tag{45}$$

which yields the lemma. $\square$

Now we can prove a minimax lower bound. Let $Y_i$ be the $i$th observed gradient, where $P_f(Y_i \mid Y_{1:i-1})$ denotes the conditional distribution of $Y_i$ under the oracle for function $f$. We have by the chain rule that

$$D_{\mathrm{kl}} \left( P_f^T \| P_g^T \right) = \sum_{i=1}^{T} \mathbb{E}_f \left[ D_{\mathrm{kl}} \left( P_f(Y_i \mid Y_{1:i-1}) \| P_g(Y_i \mid Y_{1:i-1}) \right) \right]. \tag{46}$$

It is no loss of generality to assume that the $i$th gradient query point $x_i$ is measureable with respect to $Y_{1:i-1}$ (this follows because if a randomized algorithm does well in expectation, there is at least one realization of its randomness that has small risk, so we can just take that realization and assume the procedure is deterministic). Using that we have a Gaussian oracle, we have

$$D_{\mathrm{kl}} \left( P_f(Y_i \mid Y_{1:i-1}) \| P_g(Y_i \mid Y_{1:i-1}) \right) = D_{\mathrm{kl}} \left( \mathsf{N}(f'(x_i), \sigma^2 I_{d \times d}) \| \mathsf{N}(g'(x_i), \sigma^2 I_{d \times d}) \right) \tag{47}$$

$$= \frac{1}{2\sigma^2} \left\| f'(x_i) - g'(x_i) \right\|^2 \leq \frac{1}{2\sigma^2} \kappa(f, g)^2. \tag{48}$$

Noting the not completely standard upper bound

$$\left\| P_f^T - P_g^T \right\|_{\mathrm{TV}} \leq 1 - \exp \left( -\frac{1}{2} D_{\mathrm{kl}} \left( P_f^T \| P_g^T \right) \right) \tag{49}$$

on the variation distance (see Tsybakov [17, Lemma 2.6]), we also have by Lemma 1 that

$$R_T(f, g) \geq \frac{d(f, g)}{4} \exp \left( -\frac{T}{4\sigma^2} \kappa(f, g)^2 \right). \tag{50}$$

Consider the collection of functions

$$\mathcal{F}_T := \left\{ g \in \mathcal{F} : \kappa(f,g)^2 \leq \frac{\sigma^2}{T} \right\}. \tag{51}$$

Certainly this collection is non-empty (it includes $f$). For any $\epsilon > 0$, there must exist some $g \in \mathcal{F}_T$ such that $d(f,g) \geq (1 - \epsilon)\omega_f(1/\sqrt{T})$. Let $g_T$ denote such a $g$. Then we have

$$R_T(f) \geq R_T(f, g_T) \geq \frac{d(f, g_T)}{4} e^{-\frac{1}{4}} \geq \frac{1 - \epsilon}{4} e^{-\frac{1}{4}} \omega_f\left(\frac{\sigma}{\sqrt{T}}\right). \tag{52}$$

We have

$$R_T(f) \geq \frac{1}{4e^{1/4}} \omega_f\left(\frac{\sigma}{\sqrt{T}}\right) \geq \frac{3}{16} \omega_f\left(\frac{\sigma}{\sqrt{T}}\right). \tag{53}$$

## B.2 Upper bound

Suppose that we have two functions $f_{-1}, f_1 \in \mathcal{F}$. Let

$$x^\dagger = \arg\max_{x \in \mathcal{C}} \left\{ \|f'_{-1}(x) - f'_1(x)\| \right\} \tag{54}$$

be the point at which the two functions differ the most in terms of the subgradients. Let $\theta \in \{-1, 1\}$ be the parameter. Consider an algorithm that queries the oracle with $x^\dagger$ for $T$ times. Let $Z_t$ be the response from the oracle at time $t$. Let

$$W = \frac{1}{\sqrt{T}} \sum_{t=1}^{T} Z_t - \frac{\sqrt{T}}{2}(f'_1(x^\dagger) + f'_{-1}(x^\dagger)) \tag{55}$$

With the normality assumption on the noise, we have

$$W \sim N(\theta\gamma_T, \sigma^2 I) \tag{56}$$

where

$$\gamma_T = \frac{\sqrt{T}}{2}\left(f'_1(x^\dagger) - f'_{-1}(x^\dagger)\right). \tag{57}$$

Then we construct

$$\overline{W} = \|\gamma_T\|^{-1}\gamma_T^{\mathrm{T}}W \sim N(\theta\|\gamma_T\|, \sigma^2), \tag{58}$$

which is a sufficient statistic for the problem of estimating $\theta$. Based on $\overline{W}$ we can obtain an estimate $\widehat{\theta}$ of $\theta$, and let the output of our algorithm be

$$\widehat{x}_T = \frac{x_1^* + x_{-1}^*}{2} + \widehat{\theta}\frac{x_1^* - x_{-1}^*}{2} \tag{59}$$

where $x_1^* \in \mathcal{X}_{f_1}^*$ and $x_{-1}^* \in \mathcal{X}_{f_{-1}}^*$ satisfy $\|x_1 - x_{-1}\| = \inf_{x \in \mathcal{X}_{f_1}^*} \inf_{y \in \mathcal{X}_{f_{-1}}^*} \|x - y\|$. It then follows

$$\inf_{A \in \mathcal{A}_T} \max_{\theta = \pm 1} \mathbb{E}_\theta \|\widehat{x}_A - x_\theta^*\| \leq \max_{\theta = \pm 1} \mathbb{E}_\theta \|\widehat{x}_T - x_\theta^*\| \tag{60}$$

$$\leq \frac{1}{2}\|x_1^* - x_{-1}^*\| \inf_{\widehat{\theta}} \sup_{\theta = \pm 1} \mathbb{E}_\theta |\widehat{\theta} - \theta| \tag{61}$$

$$= \frac{1}{2}\|x_1^* - x_{-1}^*\|\|\gamma_T\|^{-1}\lambda(\|\gamma_T\|, \sigma) \tag{62}$$

where $\lambda(\tau, \sigma) = \inf_{\widehat{\mu}} \sup_{\mu = \pm \tau} \mathbb{E}_\mu |\widehat{\mu} - \mu|$ is the minimax ($\ell_1$) risk of estimating the mean of $Z \sim N(\tau, \sigma^2)$ for the class $\mu \in \{-\tau, \tau\}$.

Now take $f_{-1} = f$ and $f_1 = g$. Note that $\|\gamma_T\| = \frac{\sqrt{T}}{2}\kappa(f, g)$. From (62) we have

$$R_T(f; \mathcal{F}) = \sup_{g \in \mathcal{F}} \inf_{A \in \mathcal{A}_T} \max\left\{ \mathbb{E}_f \|\widehat{x}_T - x_f^*\|, \mathbb{E}_g \|\widehat{x}_T - x_g^*\| \right\} \tag{63}$$

$$\leq \frac{1}{2} \sup_{\|\gamma_T\|} \sup_{g \in \mathcal{F}: \kappa(f,g) = \frac{2\|\gamma_T\|}{\sqrt{T}}} \|x_f^* - x_g^*\|\|\gamma_T\|^{-1}\lambda(\|\gamma_T\|, \sigma) \tag{64}$$

$$\leq \frac{1}{2} \sup_{\tau} \omega_f\left(\frac{2\tau}{\sqrt{T}}\right)\tau^{-1}\lambda(\tau, \sigma). \tag{65}$$

We have the following bound derived from [4]

$$\lambda(\tau, \sigma) \leq \tau \exp\left(-\frac{\tau^2}{4\sigma^2}\right), \tag{66}$$

which yields

$$R_T(f; \mathcal{F}) \leq \frac{1}{2} \sup_\tau \omega\left(\frac{2\tau}{\sqrt{T}}\right) \exp\left(-\frac{\tau^2}{4\sigma^2}\right). \tag{67}$$

To upper bound the last quantity, we write

$$\sup_\tau \omega\left(\frac{2\tau}{\sqrt{T}}\right) \exp\left(-\frac{\tau^2}{4\sigma^2}\right) \leq \max\left\{ \sup_{\tau \leq r} \psi(\tau), \sup_{r < \tau \leq \frac{1}{2}\epsilon_0 \sqrt{T}} \psi(\tau), \sup_{\tau > \frac{1}{2}\epsilon_0 \sqrt{T}} \psi(\tau) \right\} \tag{68}$$

for some $r > 0$, where $\psi(\tau) = \omega\left(\frac{2\tau}{\sqrt{T}}\right) \exp\left(-\frac{\tau^2}{4\sigma^2}\right)$. We bound the three terms separately by

$$\sup_{\tau \leq r} \omega\left(\frac{2\tau}{\sqrt{T}}\right) \exp\left(-\frac{\tau^2}{4\sigma^2}\right) \leq \omega\left(\frac{2r}{\sqrt{T}}\right), \tag{69}$$

and

$$\sup_{r < \tau \leq \frac{1}{2}\epsilon_0 \sqrt{T}} \omega\left(\frac{2\tau}{\sqrt{T}}\right) \exp\left(-\frac{\tau^2}{4\sigma^2}\right) \tag{70}$$

$$= \sup_{s \geq 1 \,\&\, \frac{2sr}{\sqrt{T}} \leq \epsilon_0} \omega\left(\frac{2sr}{\sqrt{T}}\right) \exp\left(-\frac{s^2 r^2}{4\sigma^2}\right) \tag{71}$$

$$\leq \sup_{s \geq 1} s^\alpha \omega\left(\frac{2r}{\sqrt{T}}\right) \exp\left(-\frac{s^2 r^2}{4\sigma^2}\right) \tag{72}$$

$$\leq \left(\frac{\sqrt{2\alpha}\sigma}{r}\right)^\alpha \omega\left(\frac{2r}{\sqrt{T}}\right) \tag{73}$$

since $\omega_f$ satisfies $\omega_f(c\epsilon) \leq c^\alpha \omega_f(\epsilon)$ for $c > 1$, $c\epsilon \leq \epsilon_0$ and some $\alpha > 0$, and

$$\sup_{\tau > \frac{1}{2}\epsilon_0 \sqrt{T}} \omega\left(\frac{2\tau}{\sqrt{T}}\right) \exp\left(-\frac{\tau^2}{4\sigma^2}\right) \leq \operatorname{diam}(\mathcal{C}) \exp\left(-\frac{\epsilon_0^2 T}{16\sigma^2}\right) \tag{74}$$

Setting $r = \sigma/2$ and noting that $\omega_f(\epsilon) \geq \epsilon^\alpha \frac{\omega_f(\epsilon_0)}{\epsilon_0^\alpha}$, we have that there exists $T_0 > 0$ such that for all $T \geq T_0$

$$R_T(f; \mathcal{F}) \leq C \omega_f\left(\frac{\sigma}{\sqrt{T}}\right) \tag{75}$$

where $C = \frac{1}{2} \max\{1, (8\alpha)^{\frac{\alpha}{2}}\}$.

## C  Proofs for superefficiency results

We begin by recalling the following results about properties of the subdifferential of a convex function $f$ and its Fenchel conjugate

$$f^*(y) := \sup_x \left\{y^T x - f(x)\right\}, \tag{76}$$

including duality between the subdifferential sets $\partial f$ and $\partial f^*$, increasing gradients, and continuous differentiability.

**Lemma 2** (Hiriart-Urruty and Lemaréchal [7]). *Let $f$ be a closed convex function. Then*

$$x \in \partial f^*(y) \quad \text{if and only if} \quad y \in \partial f(x). \tag{77}$$

*Additionally, subgradient sets are increasing in the sense that*

$$s_1 \in \partial f(x_1) \text{ and } s_2 \in \partial f(x_2) \quad \text{implies} \quad \langle s_1 - s_2, x_1 - x_2 \rangle \geq 0. \tag{78}$$

*Lastly, if $f : \mathbb{R} \to \mathbb{R}$ is strictly convex on an interval $[x_l, x_r]$, then $f^*$ is continuously differentiable on the interval $[\inf\{s : s \in \partial f(x_l)\}, \sup\{s : s \in \partial f(x_r)\}]$.*

## C.1 Moduli of continuity

**Lemma 3.** *Let $f : \mathbb{R} \to \mathbb{R}$ be a subdifferentiable convex function. Define $f_\epsilon(x) = f(x) + \epsilon x$. Then*

$$\arg\min_x f_\epsilon(x) = \partial f^*(-\epsilon) \tag{79}$$

*Moreover,*

$$\operatorname{dist}(\partial f^*(0), \partial f^*(\epsilon)) \vee \operatorname{dist}(\partial f^*(0), \partial f^*(-\epsilon)) \leq \omega_f(\epsilon) \tag{80}$$

$$\omega_f(\epsilon) \leq \sup_x\{\operatorname{dist}(x, \partial f^*(0)) : x \in \partial f^*(\epsilon)\} \vee \sup_x\{\operatorname{dist}(x, \partial f^*(0)) : x \in \partial f^*(-\epsilon)\} \tag{81}$$

*In particular, if $x_0 = \arg\min_x f(x)$ is unique and $f$ is strictly convex in a neighborhood of $x_0$, then there exists an $\epsilon_0 > 0$ such that $\epsilon \leq \epsilon_0$ implies that*

$$\omega_f(\epsilon) = \max\left\{|f^{*\prime}(\epsilon) - x_0|, |f^{*\prime}(-\epsilon) - x_0|\right\}. \tag{82}$$

*Proof.* Let $x_0 \in \arg\min_x f(x)$. Using Lemma 2, it is clear that $\arg\min_x f(x) = \partial f^*(0)$, and more generally, that

$$\partial f^*(y) = \arg\max_x \left\{y^T x - f(x)\right\} = \arg\min_x \left\{f(x) - y^T x\right\}. \tag{83}$$

We begin by providing the lower bound on $\omega_f$. For $\epsilon > 0$, define the function $f_\epsilon(x) = f(x) + \epsilon x$. Then certainly $\kappa(f, f_\epsilon) \leq \epsilon$. Moreover, we have

$$f_\epsilon^*(y) = \sup_x\{yx - f(x) - \epsilon x\} = \sup_x\{(y - \epsilon)x - f(x)\} = f^*(y - \epsilon), \tag{84}$$

so that $\arg\min_x f_\epsilon(x) = \partial f^*(-\epsilon)$. Noting that $x_0 \in \partial f^*(0)$ and that subgradients are increasing by Lemma 2, we have that

$$\arg\min_x f_\epsilon(x) = \partial f^*(-\epsilon) \leq \partial f^*(0) = \arg\min_x f(x). \tag{85}$$

That is, we have $\sup\{x_\epsilon \in \arg\min_x f_\epsilon(x)\} \leq \inf\{x_0 \in \arg\min_x f(x)\}$ and

$$\omega_f(\epsilon) \geq \inf\left\{|s_\epsilon - s_0| : s_\epsilon \in \partial f^*(-\epsilon), s_0 \in \partial f^*(0)\right\}. \tag{86}$$

An identical argument with $f_{-\epsilon}$ gives the lower bound.

For the upper bound on the modulus of continuity, we note that if $g$ is a convex function with $\kappa(f, g) \leq \epsilon$, and $x_g \in \arg\min_x g(x)$, then there must be some $s \in \partial f(x_g)$ with $\epsilon \geq s \geq -\epsilon$, because $0 \in \partial g(x_g)$, where we have used the definition of the Hausdorff distance. Now, for this particular $s$, by Lemma 2 we have that

$$x_g \in \partial f^*(s). \tag{87}$$

Again using the increasing behavior of subgradients, we obtain that

$$\inf \partial f^*(-\epsilon) \leq x_g \leq \sup \partial f^*(\epsilon), \tag{88}$$

which gives the claimed upper bound in the lemma once we recognize that $x_0 \in \partial f^*(0)$, and the definition of distance for $\omega_f$ is $d(f, g) = \inf\{|x_0 - x_g^\star| : x_0 \in \arg\min_x f(x), x_g^\star \in \arg\min_x g(x)\}$.

The final result, with the uniqueness, is an immediate consequence of the differentiability properties in Lemma 2. $\qquad\square$

Now we calculate bounds for a few example moduli of contiuity using Lemma 3. Roughly, we focus on non-pathological convex functions to allow us to give explicit calculations. Let $f : \mathbb{R} \to \mathbb{R}$ be a convex function satisfying $\partial f^*(0) = \arg\min_x f(x) = [x_l, x_r]$. In addition, assume that for $\delta > 0$, we have for some powers $k_l, k_r \geq 1$ and constants $\lambda_l > 0$ and $\lambda_r > 0$ that

$$f(x_l - \delta) = f(x_l) + \lambda_l \delta^{k_l} + o(\delta^{k_l}) \text{ and } f(x_r + \delta) = f(x_r) + \lambda_r \delta^{k_r} + o(\delta^{k_r}). \tag{89}$$

That is, in a neighborhood of the optimal region, the function $f$ grows like a polynomial. The condition (89) is not too restrictive, but does rule out functions such as $f(x) = e^{-\frac{1}{x^2}}$.

**Lemma 4.** *Let $f$ satisfy the condition* (89). *For any $c > 1$, there exists some $\epsilon_0 > 0$ such that for $\epsilon \in (0, \epsilon_0)$*

$$x_r + \left(\frac{\epsilon}{C\lambda_r k_r}\right)^{\frac{1}{k_r - 1}} \leq \inf \partial f^*(\epsilon) \leq \sup \partial f^*(\epsilon) \leq x_r + \left(\frac{C\epsilon}{\lambda_r}\right)^{\frac{1}{k_r - 1}} \tag{90a}$$

*and*

$$x_l - \left(\frac{\epsilon}{C\lambda_l k_l}\right)^{\frac{1}{k_l - 1}} \geq \sup \partial f^*(-\epsilon) \geq \inf \partial f^*(-\epsilon) \geq x_l - \left(\frac{C\epsilon}{\lambda_l}\right)^{\frac{1}{k_l - 1}}. \tag{90b}$$

*Moreover, setting $k = \max\{k_r, k_l\}$ and letting*

$$\lambda = \begin{cases} \lambda_l & \text{if } k_l > k_r, \\ \lambda_r & \text{if } k_r > k_l, \\ \max\{\lambda_r, \lambda_l\} & \text{otherwise,} \end{cases} \tag{91}$$

*we have for all $\epsilon \in (0, \epsilon_0)$ that*

$$\left(\frac{\epsilon}{C\lambda k}\right)^{\frac{1}{k-1}} \leq \omega_f(\epsilon) \leq \left(\frac{C\epsilon}{\lambda}\right)^{\frac{1}{k-1}}. \tag{92}$$

*Proof.* We focus on the right side bound (90a), as the proof of the left bound (90b) is similar. We also let the constant be $c = 2$ for simplicity.

For notational simplicity, let $\lambda = \lambda_r$ and $k = k_r$. By the fact that subgradients are increasing, we have for any $\delta > 0$ that

$$\inf \partial f(x_r + \delta) \geq \frac{f(x_r + \delta) - f(x_r)}{\delta} = \frac{\lambda\delta^k + o(\delta^k)}{\delta} = \lambda(1 - o_\delta(1))\delta^{k-1} \tag{93}$$

as $\delta \downarrow 0$. Similarly, $\delta > 0$ we have

$$\sup \partial f(x_r + \delta) \leq \frac{f(x_r + 2\delta) - f(x_r + \delta)}{\delta} = \frac{\lambda(2\delta)^k - \lambda\delta^k + o(\delta^k)}{\delta}$$
$$= \frac{\lambda k \delta^{k-1}\delta + o(\delta^k)}{\delta} = (1 + o_\delta(1))\lambda k \delta^{k-1}. \tag{94}$$

Combining inequalities (93) and (94), we thus see that there exists some $\delta_0 > 0$ such that for $\delta \in (0, \delta_0)$ we have

$$\frac{\lambda}{2}\delta^{k-1} \leq \inf \partial f(x_r + \delta) \leq \sup \partial f(x_r + \delta) \leq 2\lambda k \delta^{k-1}. \tag{95}$$

Noting that $x_r + \delta \in \partial f^*(\epsilon)$ if and only if $\epsilon \in \partial f(x_r + \delta)$ by standard subgradient calculus (recall Lemma 2), we solve for $\epsilon = \frac{\lambda}{2}\delta^{k-1}$ and $\epsilon = 2\lambda k \delta^{k-1}$ to attain inequality (90a). The bound (90b) is similar. $\square$

Lemma 4 shows that, as $\epsilon \to 0$, we have $\omega_f(\epsilon) \asymp \epsilon^{\frac{1}{k-1}}$, where $k = \max\{k_r, k_l\}$. Finally, we show a type of continuity property with the modulus of continuity.

**Lemma 5.** *Assume that $f$ has expansion* (89), *and that either (i) $k_r > k_l$ or (ii) $k_r \geq k_l$ and $\lambda_r \geq \lambda_l$. Define $g(x) = f(x) - \epsilon x$. Then for any constants $c < 1 < C$, we have*

$$\omega_g(c\epsilon) \leq (2C)^{\frac{1}{k_r - 1}} \left(\frac{\epsilon}{\lambda_r}\right)^{\frac{1}{k_r - 1}} \leq (2C^2)^{\frac{1}{k_r - 1}} e\, \omega_f(\epsilon) \tag{96}$$

*for all $\epsilon$ suitably close to $0$.*

*Proof.* We know by the increasing properties of the subgradient set and Lemma 3 that for any $c < 1$

$$\omega_g(c\epsilon) \leq \max\{\operatorname{dist}(\partial g^*(\epsilon), \partial g^*(0)), \operatorname{dist}(\partial g^*(-\epsilon), \partial g^*(0))\} \tag{97}$$
$$= \max\{\operatorname{dist}(\partial f^*(2\epsilon), \partial f^*(\epsilon)), \operatorname{dist}(\partial f^*(0), \partial f^*(\epsilon))\}, \tag{98}$$

where we have used that $g^*(y) = \sup_x\{(y+\epsilon)x - f(x)\} = f^*(y+\epsilon)$. For small enough $\epsilon > 0$, we have by Lemma 4 that

$$\sup \partial f^*(2\epsilon) \leq \left(\frac{2C\epsilon}{\lambda_r}\right)^{\frac{1}{k_r-1}}, \tag{99}$$

which gives the first inequality.

For the second inequality, we use that $\omega_f(\epsilon) \geq (\epsilon/(C\lambda_r k_r))^{\frac{1}{k_r-1}}$ to obtain

$$\left(\frac{2C\epsilon}{\lambda_r}\right)^{\frac{1}{k_r-1}} = k_r^{\frac{1}{k_r-1}}(2C^2)^{\frac{1}{k_r-1}}\left(\frac{\epsilon}{C\lambda_r k_r}\right)^{\frac{1}{k_r-1}} \leq k_r^{\frac{1}{k_r-1}}(2C^2)^{\frac{1}{k_r-1}}\omega_f(\epsilon) \leq e(2C^2)^{\frac{1}{k_r-1}}\omega_f(\epsilon) \tag{100}$$

as desired. $\square$

## C.2 Superefficiency

For distributions $P_0$ and $P_1$ define the $\chi$-divergence by

$$D_\chi(P_1\|P_0) := \int\left(\frac{dP_1}{dP_0} - 1\right)dP_1 = \int\left(\frac{dP_1}{dP_0}\right)dP_1 - 1. \tag{101}$$

The following lemma, which is a stronger version of a result due to Brown and Low [1], gives a result on superefficiency.

**Lemma 6.** *Let $\widehat{x}$ be any function of a sample $\xi$, and let $X_0$ and $X_1$ be compact convex sets (associated with distributions $P_0$ and $P_1$). Let $\text{dist}(x, X) = \inf_{y\in X}|y - x|$ and $\text{dist}(X_0, X_1) = \inf_{x_0\in X_0}\text{dist}(x_0, X_1)$. Then*

$$\mathbb{E}_{P_1}[\text{dist}(\widehat{x}, X_1)] \geq \left[\text{dist}(X_0, X_1) - \sqrt{\mathbb{E}_{P_0}[\text{dist}(\widehat{x}, X_0)^2](D_\chi(P_1\|P_0) + 1)}\right]_+ \tag{102}$$

$$\geq \text{dist}(X_0, X_1)\left[1 - \frac{\sqrt{\mathbb{E}_{P_0}[\text{dist}(\widehat{x}, X_0)^2](D_\chi(P_1\|P_0) + 1)}}{\text{dist}(X_0, X_1)}\right]_+. \tag{103}$$

*Proof.* We have

$$\mathbb{E}_{P_1}[\text{dist}(\widehat{x}, X_1)] \overset{(i)}{\geq} \text{dist}(X_0, X_1) - \mathbb{E}_{P_1}[\text{dist}(\widehat{x}, X_0)]$$

$$\overset{(ii)}{\geq} \text{dist}(X_0, X_1) - \sqrt{\mathbb{E}_{P_0}[\text{dist}(\widehat{x}, X_0)^2]\cdot\int\left(\frac{dP_1}{dP_0}\right)dP_1}$$

$$= \text{dist}(X_0, X_1) - \sqrt{\mathbb{E}_{P_0}[\text{dist}(\widehat{x}, X_0)^2](D_\chi(P_1\|P_0) + 1)}$$

where inequality (i) uses the triangle inequality and inequality (ii) uses Cauchy-Schwarz. $\square$

We now present two lemmas on $\chi$-divergence that will be useful. The first is a standard algebraic calculation.

**Lemma 7.** *Let $P_0$ and $P_1$ be normal distributions with means $\mu_0$ and $\mu_1$, respectively, and variances $\sigma^2$. Then*

$$D_\chi(P_0\|P_1) = D_\chi(P_1\|P_0) = \exp\left(\frac{(\mu_0 - \mu_1)^2}{\sigma^2}\right) - 1. \tag{104}$$

For the second lemma, we assume that $\widehat{x}$ is constructed based on noisy subgradient information from a subgradient oracle, which upon being queried at a point $x$, returns

$$f'(x) + \varepsilon, \quad\text{where } \varepsilon \overset{\text{iid}}{\sim} \mathsf{N}(0, \sigma^2) \text{ and } f'(x) = \underset{s\in\partial f(x)}{\arg\min}\{|s|\}. \tag{105}$$

The latter condition simply specifies the subgradient the oracle chooses; any specified choice of subgradient is sufficient. Because $\partial f(x)$ is a closed convex set for any $x$, we see that if $f$ and $g$ are convex functions with $\kappa(f, g) \leq \epsilon$, then $|f'(x) - g'(x)| \leq \epsilon$ with the construction (105) of subgradient oracle.

**Lemma 8.** *Let the subgradient oracle be given by* (105)*, and let $P_f^T$ and $P_g^T$ be the distributions (respectively) of the observed stochastic sub-gradients*

$$s_i = f'(x_i) + \varepsilon_i \ \ or \ \ s_i = g'(x_i) + \varepsilon_i, \tag{106}$$

*where $x_i$ is a measurable function of an independent noise variable $\xi_0$ and the preceding sequence of stochastic gradients $\{s_1, \ldots, s_{i-1}\}$. Let $\kappa(f,g) \leq \epsilon$. Then*

$$D_\chi \left( P_f^T \| P_g^T \right) \leq \exp\left( \frac{T\epsilon^2}{\sigma^2} \right) - 1. \tag{107}$$

*Proof.* Let $s_i$ be the $i$th observed stochastic subgradient in the sequence, and let the $\sigma$-field of the observed sequence through time $i$ be $\mathcal{F}_i = \sigma(\xi_0, s_1, \ldots, s_i)$. Then we have

$$D_\chi \left( P_f^T \| P_g^T \right) + 1 = \int \frac{dP_f^T(s_{1:n})}{dP_g^T(s_{1:n})} dP_f^T(s_{1:n}) \tag{108}$$

$$= \int \prod_{i=1}^{T} \left[ \frac{dP_f(s_i \mid s_{1:i-1})}{dP_g(s_i \mid s_{1:i-1})} dP_f(s_i \mid s_{1:i-1}) \right] \tag{109}$$

$$= \mathbb{E}\left[ \prod_{i=1}^{T} \mathbb{E}_{P_f}\left[ \frac{dP_f(S_i \mid \mathcal{F}_{i-1})}{dP_g(S_i \mid \mathcal{F}_{i-1})} \mid \mathcal{F}_{i-1} \right] \right]. \tag{110}$$

By the measurability assumption on $x_i$, that is, $x_i \in \mathcal{F}_{i-1}$, the inner expectation is simply one plus the $\chi^2$ distance between two distributions $\mathsf{N}(f'(x_i), \sigma^2)$ and $\mathsf{N}(g'(x_i), \sigma^2)$, which we know satisfies

$$\mathbb{E}_{P_f}\left[ \frac{dP_f(S_i \mid \mathcal{F}_{i-1})}{dP_g(S_i \mid \mathcal{F}_{i-1})} \mid \mathcal{F}_{i-1} \right] = \exp\left( \frac{(f'(x_i) - g'(x_i))^2}{\sigma^2} \right) \leq \exp\left( \frac{\epsilon^2}{\sigma^2} \right). \tag{111}$$

Taking the product over all $T$ terms yields the lemma. $\qquad\square$

**Lemma 9.** *Let $f$ be a closed convex function. Define the function*

$$H(\epsilon) := \inf\left\{ |x - x_0| : x \in \partial f^*(\epsilon), x_0 \in \partial f^*(0) \right\} \vee \inf\left\{ |x - x_0| : x \in \partial f^*(-\epsilon), x_0 \in \partial f^*(0) \right\}$$
$$= \operatorname{dist}(\partial f^*(\epsilon), \partial f^*(0)) \vee \operatorname{dist}(\partial f^*(-\epsilon), \partial f^*(0)). \tag{112}$$

*For any $0 \leq c_l < 1$ and $1 < c_u < \infty$,*

$$\omega_f(c_u \epsilon) \geq H(\epsilon) \geq \omega_f(c_l \epsilon). \tag{113}$$

**Proposition 3.** *Define $H$ to be the function* (112) *and assume additionally that $\delta < \sqrt{\frac{1}{8e}}$. If $\widehat{x}$ is any estimator such that*

$$\sqrt{\mathbb{E}_{P_f^T}\left[ \operatorname{dist}(\widehat{x}, \mathcal{X}_f^*)^2 \right]} \leq \delta \omega_f(\sigma/\sqrt{T}), \tag{114}$$

*then taking $f_1(x) = f(x) + \sqrt{\frac{\sigma^2 \log \frac{1}{8\delta^2}}{T}} x$ and $f_{-1}(x) = f(x) - \sqrt{\frac{\sigma^2 \log \frac{1}{8\delta^2}}{T}} x$, we have*

$$\max_{g \in \{f_1, f_{-1}\}} \mathbb{E}_{P_g^T}\left[ \operatorname{dist}(\widehat{x}, \mathcal{X}_g^*) \right] \geq \sup_{0 < c < \log \frac{1}{8\delta^2}} \omega_f\left( \sqrt{\frac{c\sigma^2}{T}} \right) \left( 1 - \frac{\omega_f(\sigma/\sqrt{T})}{2\sqrt{2}\omega_f(\sqrt{c\sigma^2/T})} \right) \tag{115}$$

$$\geq \frac{4 - \sqrt{2}}{4} H\left( \sqrt{\frac{\sigma^2 \log \frac{1}{8\delta^2}}{T}} \right). \tag{116}$$

*Proof.* Without loss of generality, we assume that $0 \in \arg\min_x f(x) = \partial f^*(0)$, and set $x_0 = 0$ for simplicity in the derivation. For any $\epsilon \in \mathbb{R}$, we may construct the function $f_\epsilon(x) = f(x) + \epsilon x$. Lemma 6 and Lemma 8 thus yield that that for $\mathcal{X}_\epsilon = \arg\min_x f_\epsilon(x)$, we have

$$\mathbb{E}_{P_{f_\epsilon}^T}[\operatorname{dist}(\widehat{x}, \mathcal{X}_\epsilon)] \geq \operatorname{dist}(\partial f^*(-\epsilon), \partial f^*(0)) \left[ 1 - \frac{\omega_f(\sigma/\sqrt{T})\sqrt{\delta \exp(\frac{T\epsilon^2}{\sigma^2})}}{\operatorname{dist}(\partial f^*(-\epsilon), \partial f^*(0))} \right]_+ \tag{117}$$

and

$$\mathbb{E}_{P_{f-\epsilon}^T}\left[\text{dist}(\widehat{x}, \mathcal{X}_{-\epsilon})\right] \geq \text{dist}(\partial f^*(\epsilon), \partial f^*(0))\left[1 - \frac{\omega_f(\sigma/\sqrt{T})\sqrt{\delta\exp(\frac{T\epsilon^2}{\sigma^2})}}{\text{dist}(\partial f^*(\epsilon), \partial f^*(0))}\right]_+. \quad (118)$$

In particular, with $H(\epsilon) = \text{dist}(\partial f^*(\epsilon), \partial f^*(0)) \vee \text{dist}(\partial f^*(\epsilon), \partial f^*(0))$, we have

$$\max_{g\in f_\epsilon, f_{-\epsilon}} \mathbb{E}_{P_g^T}\left[\text{dist}(\widehat{x}, \mathcal{X}_g^*)\right] \geq H(\epsilon)\left[1 - \frac{\omega_f(\sigma/\sqrt{T})\sqrt{\delta\exp(\frac{n\epsilon^2}{\sigma^2})}}{H(\epsilon)}\right]_+. \quad (119)$$

Take $\epsilon^2 = \frac{\sigma^2}{T}\log\frac{1}{8\delta^2}$ to obtain

$$\max_{g\in f_\epsilon, f_{-\epsilon}} \mathbb{E}_{P_g^T}\left[\text{dist}(\widehat{x}, \mathcal{X}_g^*)\right] \geq H\left(\sqrt{\frac{\sigma^2\log\frac{1}{8\delta^2}}{T}}\right)\left[1 - \frac{\omega_f(\sigma/\sqrt{T})}{2\sqrt{2}H(\sigma\log^{\frac{1}{2}}\frac{1}{8\delta^2}/\sqrt{T})}\right]_+. \quad (120)$$

Notably, by Lemma 3, our w.l.o.g. assumption and the fact that subgradients are increasing, we have that for any constant $(\log\frac{1}{8\delta^2})^{-\frac{1}{2}} \leq c < 1$ that

$$\omega_f\left(\frac{\sigma}{\sqrt{T}}\right) \leq \omega_f\left(c\sqrt{\frac{\sigma^2\log\frac{1}{8\delta^2}}{T}}\right) \quad (121)$$

$$\leq \sup\left\{\text{dist}(x, X_0) : x \in \partial f^*\left(c\frac{\sigma\log^{\frac{1}{2}}\frac{1}{8\delta^2}}{\sqrt{T}}\right)\right\} \vee \sup\left\{\text{dist}(x, X_0) : x \in \partial f^*\left(-c\frac{\sigma\log^{\frac{1}{2}}\frac{1}{8\delta^2}}{\sqrt{T}}\right)\right\} \quad (122)$$

$$\leq \sup\left\{\text{dist}(x, X_0) : x \in \partial f^*\left(\frac{\sigma\log^{\frac{1}{2}}\frac{1}{8\delta^2}}{\sqrt{T}}\right)\right\} \vee \sup\left\{\text{dist}(x, X_0) : x \in \partial f^*\left(-\frac{\sigma\log^{\frac{1}{2}}\frac{1}{8\delta^2}}{\sqrt{T}}\right)\right\} \quad (123)$$

$$= H\left(\frac{\sigma\log^{\frac{1}{2}}\frac{1}{8\delta^2}}{\sqrt{T}}\right). \quad (124)$$

In particular, we have the lower bound

$$\max_{g\in f_\epsilon, f_{-\epsilon}} \mathbb{E}_{P_g^T}\left[\text{dist}(\widehat{x}, \mathcal{X}_g^*)\right] \geq H\left(\sqrt{\frac{\sigma^2\log\frac{1}{8\delta^2}}{T}}\right)\frac{4-\sqrt{2}}{4}. \quad (125)$$

This is the desired result. $\qquad\square$

Proposition 3 is a basic result on superefficiency that we may specialize to obtain more concrete results. We would like give a result that holds when $f^*$ is differentiable in a neighborhood of 0, which is equivalent to $f$ being strictly convex in a neighborhood of $x_0 = \arg\min_x f(x)$, by Lemma 2. This would mean that the function $H$ defined in Proposition 3 satisfies

$$H(\epsilon) = \max\{|f^{*\prime}(\epsilon) - x_0|, |f^{*\prime}(-\epsilon) - x_0|\} = \omega_f(\epsilon) \quad (126)$$

for all small enough $\epsilon > 0$. In this setting, we obtain

**Corollary 2.** *Let the conditions of Proposition 3 hold, and let $f$ be strictly convex in a neighborhood of $x_0 = \arg\min_x f(x)$. Assume that $\widehat{x}$ is any estimator satisfying*

$$\sqrt{\mathbb{E}_{P_f^T}[(\widehat{x} - x_0)^2]} \leq \delta\omega_f(\sigma/\sqrt{T}), \quad (127)$$

*where $\delta < \sqrt{\frac{1}{8e}}$. Define $f_{\pm 1}(x) = f(x) \pm \sqrt{\frac{\sigma^2\log\frac{1}{8\delta^2}}{T}}x$. Then for large enough $T$,*

$$\max_{g\in\{f_1, f_{-1}\}} \mathbb{E}_{P_g^T}|\widehat{x} - x_g^\star| \geq \frac{4-\sqrt{2}}{4}\omega_f\left(\sqrt{\frac{\sigma^2\log\frac{1}{8\delta^2}}{T}}\right). \quad (128)$$

This corollary has a striking weakness, however—the right hand side depends on $\omega_f$, rather than $\omega_g$, which is what we would prefer. We can, however, state a simpler result that is achievable.

**Corollary 3.** *Let $f$ be any convex function satisfying the asymptotic expansion (89) around its optimum. Suppose that $\widehat{x}$ is any estimator such that*

$$\sqrt{\mathbb{E}_{P_f^T}[\text{dist}(\widehat{x}, \mathcal{X}_f^*)^2]} \leq \delta \omega_f \left( \frac{\sigma}{\sqrt{T}} \right), \tag{129}$$

*where $\delta < \sqrt{\frac{1}{8e}}$. Define $g_{-1}(x) = f(x) - \epsilon_T x$ and $g_1(x) = f(x) + \epsilon_T x$, where $\epsilon_T = \sqrt{\frac{\sigma^2 \log \frac{1}{8\delta^2}}{T}}$, and let $k = k_r \vee k_l$. Let $C > 1$ and $0 < c < 1$ be otherwise arbitrary numerical constants. Then for one of $g \in \{g_{-1}, g_1\}$, there exists $T_0$ such that $T \geq T_0$ implies*

$$\mathbb{E}_{P_g} \left[ \text{dist}(\widehat{x}, \mathcal{X}_g^*) \right] \geq \frac{4 - \sqrt{2}}{4(2C^2)^{\frac{1}{k-1}} e} \omega_g \left( c \sqrt{\frac{\sigma^2 \log \frac{1}{8\delta^2}}{T}} \right). \tag{130}$$

*Proof.* Without loss of generality, we assume that $k_r \geq k_l$, and if $k_l = k_r$ then $\lambda_r \geq \lambda_l$. By inspection of the proof of Proposition 3, we have that

$$\mathbb{E}_{P_{g_{-1}}^T}[\text{dist}(\widehat{x}, \partial g_{-1}^*(0))] \geq \frac{1}{2} \text{dist}(\partial f^*(\epsilon_T), \partial f^*(0)). \tag{131}$$

Moreover, we know that for suitably large $n$, we have by Lemma 4

$$\text{dist}(\partial f^*(\epsilon_T), \partial f^*(0)) = \text{dist}(\partial f^*(\epsilon_T), \partial f^*(0)) \vee \text{dist}(\partial f^*(-\epsilon_T), \partial f^*(0)) \tag{132}$$
$$\geq \omega_f(c\epsilon_T) \tag{133}$$

for any $c < 1$. Then Lemma 5 implies that for any $C > 1$, there exists $T_0$ such that $T \geq T_0$ implies

$$\omega_f(c\epsilon_T) \geq \frac{1}{(2C^2)^{\frac{1}{k-1}} e} \omega_{g_{-1}}(c^2 \epsilon_T). \tag{134}$$

This gives the desired result. $\square$

As an immediate consequence of Corollary 3, we see that if there exists any sequence $\delta_T \to 0$ with $\liminf_T e^T \delta_T = \infty$ such that

$$\sqrt{\mathbb{E}_{P_f} \left[ \text{dist}(\widehat{x}, \mathcal{X}_f^*)^2 \right]} \leq \delta_T \omega_f \left( \frac{\sigma}{\sqrt{T}} \right), \tag{135}$$

then there exists a sequence of convex functions $g_T$, with $\kappa(f, g_T) \to 0$, such that

$$\liminf_T \frac{\mathbb{E}_{P_{g_T}} \left[ \text{dist}(\widehat{x}, \mathcal{X}_{g_T}) \right]}{\omega_{g_T} \left( \sqrt{\frac{\sigma^2 \log \delta_T^{-1}}{T}} \right)} > 0. \tag{136}$$

# D   Algorithm

## D.1   Proof of Proposition 2

First, by the monotonicity of the derivative $f'$, note that the interval $\mathcal{I}_\delta$ is such that $x \in \mathcal{I}_\delta$ holds if and only if $|f'(x)| < C_\delta/\sqrt{T_0}$. Now suppose that at round $e$, $(a_e, b_e) \cap \mathcal{I}_\delta \neq \emptyset$. For the next round, if $x_e = (a_e + b_e)/2 \in \mathcal{I}_\delta$, then $(a_{e+1}, b_{e+1}) \cap \mathcal{I}_\delta \neq \emptyset$. Otherwise, if $x_e \notin \mathcal{I}_\delta$, we know that $|f'(x_e)| \geq C_\delta/\sqrt{T_0}$, and without loss of generality, we assume that it is positive. Then, we have

$$\mathbb{P}\left((a_{e+1}, b_{e+1}) \cap \mathcal{I}_\delta \neq \emptyset\right) = \mathbb{P}\left(\mathsf{N}\left(f'(x_e), \frac{\sigma^2}{T_0}\right) < 0\right) = \mathbb{P}\left(\mathsf{N}(0,1) > \frac{\sqrt{T_0} f'(x_e)}{\sigma}\right) \tag{137}$$

$$\leq \mathbb{P}\left(\mathsf{N}(0,1) > \frac{C_\delta}{\sigma}\right) \leq \frac{\sigma}{C_\delta \sqrt{2\pi}} \exp\left(-\frac{C_\delta^2}{2\sigma^2}\right) \tag{138}$$

Therefore,

$$\mathbb{P}\left((a_{e+1}, b_{e+1}) \cap \mathcal{I}_\delta \neq \emptyset \big| (a_e, b_e) \cap \mathcal{I}_\delta \neq \emptyset\right) \geq 1 - \frac{\sigma}{C_\delta \sqrt{2\pi}} \exp\left(-\frac{C_\delta^2}{2\sigma^2}\right) \tag{139}$$

It then follows that

$$\mathbb{P}\left((a_E, b_E) \cap \mathcal{I}_\delta \neq \emptyset\right) = \mathbb{P}\left((a_e, b_e) \cap \mathcal{I}_\delta \neq \emptyset \text{ for } e = 1, \dots, E\right) \tag{140}$$

$$= \prod_{e=0}^{E-1} \mathbb{P}\left((a_{e+1}, b_{e+1}) \cap \mathcal{I}_\delta \neq \emptyset \big| (a_e, b_e) \cap \mathcal{I}_\delta \neq \emptyset\right) \tag{141}$$

$$\geq \left(1 - \frac{\sigma}{C_\delta \sqrt{2\pi}} \exp\left(-\frac{C_\delta^2}{2\sigma^2}\right)\right)^E \tag{142}$$

$$\geq 1 - \frac{E\sigma}{C_\delta \sqrt{2\pi}} \exp\left(-\frac{C_\delta^2}{2\sigma^2}\right) \tag{143}$$

$$\geq 1 - \delta \tag{144}$$

by the choice of $C_\delta$.

## D.2 Proof of Corollary 1

By the polynomial growth condition, we have for $T > \sigma^2/\epsilon_0$,

$$\omega_f(\epsilon_0) \leq \left(\frac{\epsilon_0 \sqrt{T}}{\sigma}\right)^\alpha \omega_f\left(\frac{\sigma}{\sqrt{T}}\right). \tag{145}$$

Since $r = \frac{1}{2}\alpha_0 \geq \frac{1}{2}\alpha$ and $E = \lfloor r \log T \rfloor$,

$$2^{-E}(b_0 - a_0) \leq 2(b_0 - a_0)T^{-r} \leq 2(b_0 - a_0)T^{-\frac{1}{2}\alpha} \leq \frac{2(b_0 - a_0)\epsilon_0^\alpha}{\omega_f(\epsilon_0)\sigma^\alpha} \omega_f\left(\frac{\sigma}{\sqrt{T}}\right) \tag{146}$$

By the expression we obtained in Example 1,

$$\sup\{\inf_{x \in \mathcal{X}_f^*} |x - y| : y \in \mathcal{I}_\delta\} \tag{147}$$

$$= \omega_f\left(\frac{C_\delta}{\sqrt{T_0}}\right) \leq \left(\sqrt{2r\left(\log(r \log T) + \log \frac{1}{\delta}\right) \log T}\right)^\alpha \omega_f\left(\frac{\sigma}{\sqrt{T}}\right) \tag{148}$$

for $T$ large enough. Therefore, we obtain that there exist $T' > 0$ such that for $T > T'$,

$$\inf_{x \in \mathcal{X}_f^*} |x_E - x| \leq \widetilde{C} \omega_f\left(\frac{1}{\sqrt{T}}\right) \tag{149}$$

where

$$\widetilde{C} = \frac{2(b_0 - a_0)\epsilon_0^\alpha}{\omega_f(\epsilon_0)\sigma^\alpha} + \left(\sqrt{2r\left(\log(r \log T) + \log \frac{1}{\delta}\right) \log T}\right)^\alpha. \tag{150}$$