[Reviews · NeurIPS 2016]

Reviewer 1

Summary

The authors advocate the interest risk bounds for convex stochastic optimization obtained on 2-point subproblems.

Qualitative Assessment

While I would certainly agree with the claim that developing local minimax bounds is of interest, I feel that the interest of considering risks for 2-point subproblems is unclear. Note that many (probably, a majority of) known bounds in convex stochastic optimization are obtained on 2-point subproblems (see, e.g. Nemirovski, Yudin book, “Fabian-type” bounds [Nazin 89] for smooth optimization, etc). In other words, the minimax risks (up to an absolute factor) for general classes of convex problems are attained already on the hardest 2-point subproblems (1-parametric families), so that, in the paper notation, R_T(\cal F)\sim \sup_{f\in \cal F} R_T(f, \cal F) On the other hand, it is unclear for me if the notion of R_T(f, \cal F) is of interest for its own sake (not as just a technical tool to construct a more general lower bound). For instance, it is not clear why and when the “2-point risk” R_T(f, \cal F) is an adequate measure of local complexity. Note that this is generally not the case in nonparametric estimation (except for few situations such as the case of linear functional estimation in [4-6]). Note also that the authors’ Remark 2 points in the same direction. Nevertheless, the paper contains some interesting results (e.g., Proposition 1) and I consider it above the acceptance threshold. I have also few minor comments: 1) Please mention that \|\cdot\| stands for the Euclidean norm 2) L. 92 The choice of the error measure err(x,f) (same in l. 121) seems to be peculiar for convex functions. 3) What are the examples of functions satisfying (12) except for those “uniformly convex” in the vicinity of the optimum? 4) Please, mention references for the existing lower bounds in the examples 1-3 (e.g., [Nazin 89] for example 3, etc). 5) L.164: should be (18) instead of (21). 6) L. 258: Note that algorithms proposed in [8] are clearly applicable in the situation when condition as in l. 232 is verified. Furthermore, the log factor appears only in the adaptive version of those algorithms. Note also the correct reference to [8]: Juditsky, A., & Nesterov, Y. (2014). Deterministic and stochastic primal-dual subgradient algorithms for uniformly convex minimization. Stochastic Systems,4, 1-37 (see also https://hal.archives-ouvertes.fr/hal-00508933, 2010) Citation [Nazin 89]: NAZIN, A. (1989). Information inequalities in stochastic gradient optimization and optimal realizable algorithms Automation & Remote Control, 50:4, 531–540.

Confidence in this Review

3-Expert (read the paper in detail, know the area, quite certain of my opinion)


Reviewer 2

Summary

This paper attempts to derive a notion of complexity for stochastic gradient descent that gets away from the worst-case bounds on Yudin and Nemirovski. The idea is that specific functions to be optimized may have structure that allows for significantly better performance of SGD, compared to the worst-case function in the appropriate class as defined by Y & N (namely, Lipschitz, or Strongly Convex). The formulation given is: R_T(f;\mathcal{F}) = \sup_{g \in F} inf_{A \in A_T} max_{h \in {f,g}}: E[err(\hat{x}_A,h)] This allows the authors to define a notion of a modulus of continuity for a class of functions \mathcal{F}, which is the biggest distance between minimizers given that sub gradients are all \epsilon-close. They then consider how the modulus scales, and say that it has polynomial growth if scaling epsilon by c gives a c^{\alpha} scaling of the modulus (for some alpha > 0). Basically this says that the steeper a function is around its optimum, the better the rate of convergence, and conversely. In terms of the writing of the paper, it would be useful to clarify definition (4), where it appears that the algorithm A is allowed to depend explicitly on the function f and the selected worst-case nearby function g. I found only one typo — on line 188.5, “…half of the interval if the estimated sign is negative”

Qualitative Assessment

It is interesting to try to get away from the worst-case analysis that has characterized our thinking of SGD. And the super efficiency result is interesting. But this paper seems preliminary in many ways. The noise model for the stochastic gradients is limited. The algorithmic side is not developed, and other results, too, seem too limited. In fairness, the authors essentially point all of this out in their conclusion. Still, it seems that while the concept of a local modulus of continuity is interesting as is the overall agenda, that the concrete and usable results are limited, at least in the paper’s current form.

Confidence in this Review

2-Confident (read it all; understood it all reasonably well)


Reviewer 3

Summary

The authors introduce a localized form of minimax complexity for individual functions. The traditional minimax complexity for convex optimization measures the hardness of optimizing all possible functions in a given function class. By contrast, the proposed local minimax complexity with respect to a particular function, and a function class measures the hardness of optimizing the worst alternative to that function, where the alternative is chosen from the function class of interest. The main result of the paper, Theorem 1, reveals that the local minimax complexity for a function can be upper- and lower-bounded, up to constant factors, by the modulus of continuity (MoC) of that function with respect to the function class of interest. Theorem 2 provides another operational interpretation of the MoC of a function: if an algorithm outperforms the MoC of a function, then it performs worse than the MoC of a nearby function. The authors also propose a universal algorithm that can perform close to the MoC of any function.

Qualitative Assessment

This paper proposes an interesting definition of local minimax complexity and reveals its connection to the modulus of continuity of a function. Some technical comments: 1. It would be better to explicitly state how the algorithm works when defining the local minimax complexity. In the definition of the traditional minimax complexity Eq. (3) and the definition of the local minimax complexity Eq. (4), the knowledge of the algorithms are different: in the former case, the algorithm knows a-priori only the entire function set F; while in the latter case, the algorithm knows that the function set is {f,g}, meaning that different algorithms can be used for different g’s. The authors should state this difference in the problem formulation to avoid confusion. 2. Both the definitions of local minimax complexity and the MoC of a function are wrt to some function set F. However, in the paper only the case where F is the set of Lipschitz convex functions is considered. It would be nice to see some general results or examples that reveal how these quantities depend on the properties of more restricted F. 3. In Example 1, an expression of MoC is given for the one-dimensional convex function, but no explanation is given why this is the case. 4. All results in the paper are based on the assumption that the oracle's output is disturbed by the Gaussian noise. No results are given for general noisy oracles.

Confidence in this Review

2-Confident (read it all; understood it all reasonably well)


Reviewer 4

Summary

The paper profives a plausible definition of local minimax complexity of optimization which goes beyond the traditional notion of optimization and considers the complexity of separating a function from its local class. The notion is interesting, the authors provide adaptive algorithms for the same and demonstrate some proof of concept experiments, albeit for one dimensional data, using binary search.

Qualitative Assessment

The main drawback is that no analogue is provided for more than 1 dimension. This is the major reason for not giving the manuscript a higher rating. Many different techniques work for one dimensions, for many spurious reasons. That being said the paper has promise and can lead to more research. The main contribution of this paper would be to restart conversations of complexity measures which are alternate to the worst case (some examples already exist, and are reviewed in the manuscript, but more is helpful).

Confidence in this Review

2-Confident (read it all; understood it all reasonably well)


Reviewer 5

Summary

The paper introduces the notion of local minimax complexity for stochastic convex optimization (SCO). The setting they consider is very similar to the classical notion of (first order) SCO: one wishes to minimize a unknown function f from some family F, and one may repeatedly query an oracle with a point x, and the oracle returns a subgradient of f at x with Gaussian noise (they have an additional assumption that the oracle always returns the same subgradient, but this is not a big deal). The traditional goal in this setting is then to approximately minimize f using as few oracle calls as possible. The traditional minimax risk is max over f in F of min over algorithms A of the expected error that A makes on f. That is, the algorithm needs to work to minimize all f in the class. In contrast, in this paper, they fix a function f and they define the local minimax complexity of optimizing f the max over g in F of min over algorithms A of the expected error A makes on either g or f. That is, the algorithm needs only differentiate between f and g (and these are known to the algorithm). In some sense, this captures a notion of "instance-by-instance" optimality for SCO. The paper then shows that when F is the set of Lipschitz convex functions, the local minimax risk is exactly captured by a notion of modulus of continuity that they define. However, the algorithm which achieves the upper bound here is not actually useful for optimization. They also show a "superefficiency" result: that any algorithm which outperforms the local minimax risk of a function at some function must perform substantially worse on some nearby function. Finally, they give an algorithm which achieves local minimax risk for SCO in 1D.

Qualitative Assessment

This is a very interesting paper, and a novel attempt to capture a notion of per-instance optimal rates for SCO. The tight connection to the modulus of continuity they define is very nice, and so is the super-efficiency result. I think this is a very interesting new direction which may yield some nice algorithmic ideas in the future. The paper does have some drawbacks. In my opinion, the algorithms presented are not strong, although the framework is very interesting. The 1D algorithm is in my opinion not vey satisfying, and neither is the information theoretic upper bound presented. The latter is particularly worrisome because it suggests to me that perhaps this notion of local minimax complexity is not quite the right thing to do, since it allows these algorithms which cheat so much. The minimax setting itself is a bit strange, because the noise is assumed to be Gaussian, whereas usually it suffices to have a second moment condition on the gradient oracle.

Confidence in this Review

2-Confident (read it all; understood it all reasonably well)